# A novel multi-modal retrieval framework for tracking vehicles using natural language descriptions

**Changhao Zhang, Zhandong Liu** \*, **Ke Li, Yong Li, Xiangwei Qi, Nan Ding**

College of Computer Science and Technology, Xinjiang Normal University, Urumqi, China

* lzd0825@mail.ustc.edu.cn

## Abstract

Recent advances in multimodal and contrastive learning have significantly enhanced image and video retrieval capabilities. This fusion provides numerous opportunities for multi-dimensional and multi-view retrieval, especially in multi-camera surveillance scenarios in traffic environments. This paper introduces a novel Multi-modal Vehicle Retrieval (MVR) system designed to retrieve the trajectories of tracked vehicles using natural language descriptions. The MVR system integrates an end-to-end text-video comparison learning model, utilizes CLIP for feature extraction, and uses a matching control system and multi-context-based attributes. Post-processing techniques are used to eliminate erroneous information. By comprehensively understanding vehicle characteristics, the MVR system can effectively identify trajectories based on natural language descriptions. Our method achieves a mean reciprocal ranking (MRR) score of 0.8966 on the test data set of the 7$th$ AI City Challenge Track 2 for retrieving tracked vehicles through natural language descriptions, surpassing the previous top-ranked result on the public leaderboard.

## Introduction

In intelligent transportation systems, cameras are deployed for uninterrupted data collection, resulting in a continuous stream of video footage. This continuous recording leads to a large-scale candidate pool for vehicle retrieval, which is a vital component of AI-driven smart city initiatives. This continuous recording forms a large-scale candidate pool for vehicle retrieval, which is crucial for AI-driven smart city initiatives. Traditionally, vehicle retrieval systems have primarily relied on image-to-image matching, which is known as vehicle re-identification (Vehicle Re-ID). However, these methods require vehicle image queries to track the target vehicle trajectories, which may not always be feasible in real-world scenarios. To address this limitation and enhance smart city operations, the AI City Challenge introduced various traffic-related tasks, including "tracking vehicle retrieval through natural language description [1]", which has witnessed significant progress in recent years. Compared to image queries, natural language descriptions offer greater intuitiveness and easier accessibility, enabling flexible applications, such as fuzzy vehicle searches. While vision-based vehicle

**Data availability statement:** In this study, we utilized the dataset from "Challenge Track 2:

Tracked-Vehicle Retrieval by Natural Language Descriptions" of the 2023 AI CITY CHALLENGE (https://www.aicitychallenge.org/2023-challenge-tracks/). The dataset is owned by The AI City Challenge Organization Committee, and the committee's contact information is Email: aicitychallenges@gmail.com. Due to the data-sharing restrictions imposed by the dataset owner, we are unable to deposit the dataset used in this study in a public repository or provide direct access to the data. The restrictions are primarily due to the terms of use set by The AI City Challenge Organization Committee, which prohibit the redistribution of the dataset to third parties. We have adhered to these terms throughout our research. Researchers interested in accessing the dataset may contact The AI City Challenge Organization Committee directly at aicitychallenges@gmail.com to request access, in compliance with the dataset's terms of use.

**Funding:** 1) National Natural Science Foundation of China (Grant No. 62162061 and Grant No. 62262066). 2) Xinjiang Normal University Doctoral Initiation Fund Project (Grant No. XJNUBS2115). 3) Xinjiang Normal University Youth Top Talents Project (Grant No. XJNUQB2022-21). 4) Xinjiang Key Research and Development Program (2022B01007-1).

**Competing interests:** The authors have declared that no competing interests exist.

re-identification has made remarkable strides, text-based systems are gaining traction due to their ability to provide clear and concise semantic information, including speed, direction, location, color, and size. Text-image retrieval, as a cross-modal task, involves learning modality representations and their shared embedding space to capture latent features. Consequently, research in language-vision retrieval has focused on improving the learning of embedding feature vectors for higher accuracy in representation matching. Recent advancements in deep learning frameworks have shifted the focus toward uncovering semantic concepts within the language and vision inputs, rather than solely relying on matching and ranking algorithms. This approach enhances the robustness and flexibility of vehicle retrieval systems, making them more adaptable to real-world scenarios.

For instance, Zhang et al. [2] developed a multi-granularity retrieval system that ranked first in the 2022 leaderboard, achieving a mean reciprocal ranking (MRR) of 56.52%. Le et al. [3] proposed a semi-supervised domain adaptation training process, using context-sensitive post-processing methods to analyze motion and optimize retrieval results. Other notable works, such as [4], has designed a symmetric network model to learn the representation between language descriptions and vehicles, while incorporating spatial relationship modeling. Additionally, several research groups[2,4,5] have leveraged the global motion imagery framework [6], to engineer a specialized stream dedicated to vehicle motion analysis. In addition, Xie et al. [7] have proposed a framework that integrates an end-to-end text-video contrastive learning model, a CLIP-based small sample domain adaptation method, and a semi-centralized control optimization system to identify valid trajectories with natural language descriptions by fully understanding the knowledge of vehicle type, color, maneuverability, and surrounding environment. Another approach, outlined in [3] , utilizes CLIP as a baseline model and incorporates a semi-supervised domain adaptation method to bridge the domain gap between the training and the tests. Additionally, it employs multi-context post-processing technology to effectively enhance the retrieval results. Despite these advances, there is still room for improvement in the performance of text-based vehicle retrieval systems.

Building upon the success of previous research[2–7], we propose an innovative deep learning system named Multimodal Language Vehicle Retrieval (MVR), which is designed to enhance text-based vehicle retrieval capabilities. The MVR system encompasses four core components: an end-to-end text-video contrastive learning module, a vehicle motion module, a matching control system, and a multi-context pruning method. The text-video contrastive learning module plays a pivotal role by integrating video and text information to effectively extract video features. The vehicle motion module excels at identifying the motion trajectories of vehicles within the video, thereby facilitating more accurate comparison and matching with the text. The matching control system ensures that the most optimal text-video match is achieved. Furthermore, the multi-context pruning method refines the retrieval results by applying multiple contextual constraints, enabling the distinction between visually similar vehicle trajectories. Through the synergistic functioning of these core components, the MVR strategy markedly improves the accuracy of vehicle trajectory identification, thus pushing forward the progress in the field of multimodal retrieval. Therefore, the main contributions of our paper are as follows:

- We have developed an enhanced vehicle retrieval model that builds upon the CLIP architecture, incorporating the CoT block and Agent attention mechanism, as well as variants of ViT. The integration of these components not only boosts performance but also refines the alignment between visual and textual information. This approach results in more precise differentiation of visually similar vehicle trajectories.

- We refine the results through data fusion, a matching control system, and multi-contextual pruning techniques, enabling more precise identification and removal of mismatched vehicle tracks, thereby significantly enhancing the overall accuracy of the model.
- Our system achieved 1st place on the testing set of the 7*th* AI City Challenge Track 2, with a mean reciprocal ranking (MRR) score of 0.8966.

## Related work

### Vehicle re-identification

Vehicle re-identification (ReID) is crucial in computer vision for recognizing specific vehicles across multiple camera viewpoints, especially when license plate details are unavailable. This technology significantly enhances intelligent transportation systems in smart cities by facilitating the tracking of vehicle movements and detection of traffic irregularities. The main challenge is overcoming the scarcity of real-world data, which leads to discrepancies between training and test datasets, including issues like occlusion and diverse appearances across environments. Addressing this domain bias is crucial to improving the reliability and effectiveness of vehicle re-identification systems.

Recent advancements in deep learning have significantly enhanced the accuracy of vehicle re-identification tasks. In terms of network architecture, PAMTRI [8], a pose-aware multi-task learning network effectively addressing both vehicle re-identification and attribute classification simultaneously. Another framework [9] merges the strengths of ResNet and Swin Transformer to create a robust vehicle representation. Additionally, the SSBVER model employs self-supervision through self-training and knowledge distillation to boost performance during training. To bridge the gap between part and global features, a simple Triplet Contrastive Representation Learning (TCRL) framework [10] has been developed, leveraging cluster features for this purpose. Furthermore, a dual distance center loss (DDCL) [11] has been introduced to address limitations inherent in traditional center loss methodologies. To tackle appearance ambiguities in vehicles, a framework [12] has been developed that estimates camera network topology and integrates appearance with spatial-temporal similarities.For addressing domain bias, dual embedding expansion (DEx) [13] offers an embedding expansion technique that utilizes multiple networks. Moreover, Zheng et al. [14] have contributed a large-scale vehicle dataset named VehicleNet to combat the issue of limited training images. However, our task diverges from traditional vehicle re-identification by focusing on a cross-modal retrieval problem that integrates both language and vision modalities.

### Multi-modal and contrastive learning

Detection algorithms based on multimodal and contrastive learning, exemplified by CLIP [15], MoCo [16], and BLIP [17], face inherent challenges due to modality heterogeneity and data dependency. A core challenge lies in aligning diverse modalities (e.g., text, images) into a unified representation space. Temporal or spatial misalignments between modalities, such as discrepancies between facial expressions and speech timing in sentiment analysis, often result in suboptimal fusion [18]. Even state-of-the-art frameworks like CLIP require meticulously aligned image-text pairs, which are labor-intensive to curate and prone to annotation errors [19]. Furthermore, these methods heavily depend on large-scale paired datasets for training. For instance, CLIP-like models necessitate millions of image-text pairs, limiting their applicability in domains like medical imaging, where spatial transcriptomics models (e.g., mclSTExp) struggle with scarce labeled data [20]. Weakly supervised approaches, such as those for blind

image super-resolution, also rely on idealized degradation assumptions, reducing their adaptability to real-world complexities [21]. Modality imbalance exacerbates these issues; systems like LIMoE prioritize image tokens over text due to routing biases, leading to unstable representations [19]. Asymmetric feature granularity in drug-target interaction prediction further complicates multimodal fusion [22].

Another significant limitation concerns robustness and scalability. Performance deteriorates markedly under noisy or incomplete inputs. For example, smoke-obscured object detectors falter when test conditions deviate from training data [23], and models like CSK-Net struggle with missing modalities (e.g., infrared data) despite using knowledge distillation [24]. Computational overhead further constrains deployment; transformer-based architectures for gene expression prediction encounter scalability bottlenecks with high-resolution images [20], while LIMoE's mixture-of-experts design necessitates careful regularization to prevent expert collapse [19]. Generalization is another hurdle; models pretrained on natural images underperform in specialized domains like medical imaging without fine-tuning, and drug-target interaction predictors fail to generalize to novel protein families [22]. Lastly, the opacity of contrastive embeddings undermines trust in critical applications. Even interpretable frameworks like LIMoE lack transparent decision-making explanations [19], and gene expression models provide limited insights into histology-gene correlations [20]. Collectively, these challenges highlight the need for more adaptive, efficient, and interpretable multimodal frameworks.

In contrast to existing methods, the approach proposed in this study demonstrates substantial improvements across multiple dimensions. Firstly, as depicted in Fig 1, our MVR system incorporates a Vehicle Motion Module (VMM) that captures the dynamic attributes of tracked vehicles, representing a significant advancement over traditional methods that typically focus solely on static attributes. The integration of VMM has notably increased the Mean Reciprocal Rank (MRR) from 0.2562 to 0.3812, indicating a substantial enhancement in the precision of matching textual queries to relevant vehicle records.

Secondly, the introduction of the Match Control System (MCS) further enhances the system's ability to distinguish between similar vehicle trajectories. This addition has led to an MRR improvement of 0.0268, resulting in a refined MRR of 0.4080. The MCS enables our system to handle complex and similar vehicle trajectories with greater accuracy, addressing a common challenge faced by existing methods.

Lastly, our method employs a Multi-Context Pruning Method that leverages bidirectional attributes to maximize the effectiveness of our pruning strategy. This pruning technique, designed to eliminate irrelevant search candidates, has significantly boosted the MRR to an impressive peak of 0.8966 through two stages of pruning. This approach not only improves the efficiency of the retrieval process but also elevates the overall performance of our MVR model.

Moreover, our MVR system offers a more comprehensive and robust framework for semantic similarity learning compared to conventional contrastive learning methods. It effectively captures a broader spectrum of semantic relations and improves the handling of noisy or misaligned data. This holistic approach ensures that our system can accurately retrieve tracked vehicles based on natural language descriptions, overcoming the limitations of current methods.

Furthermore, the success of CLIP can be partly attributed to its adoption of the Transformer architecture, which efficiently processes sequence data and provides robust contextual understanding capabilities. The Transformer architecture excels in contrastive

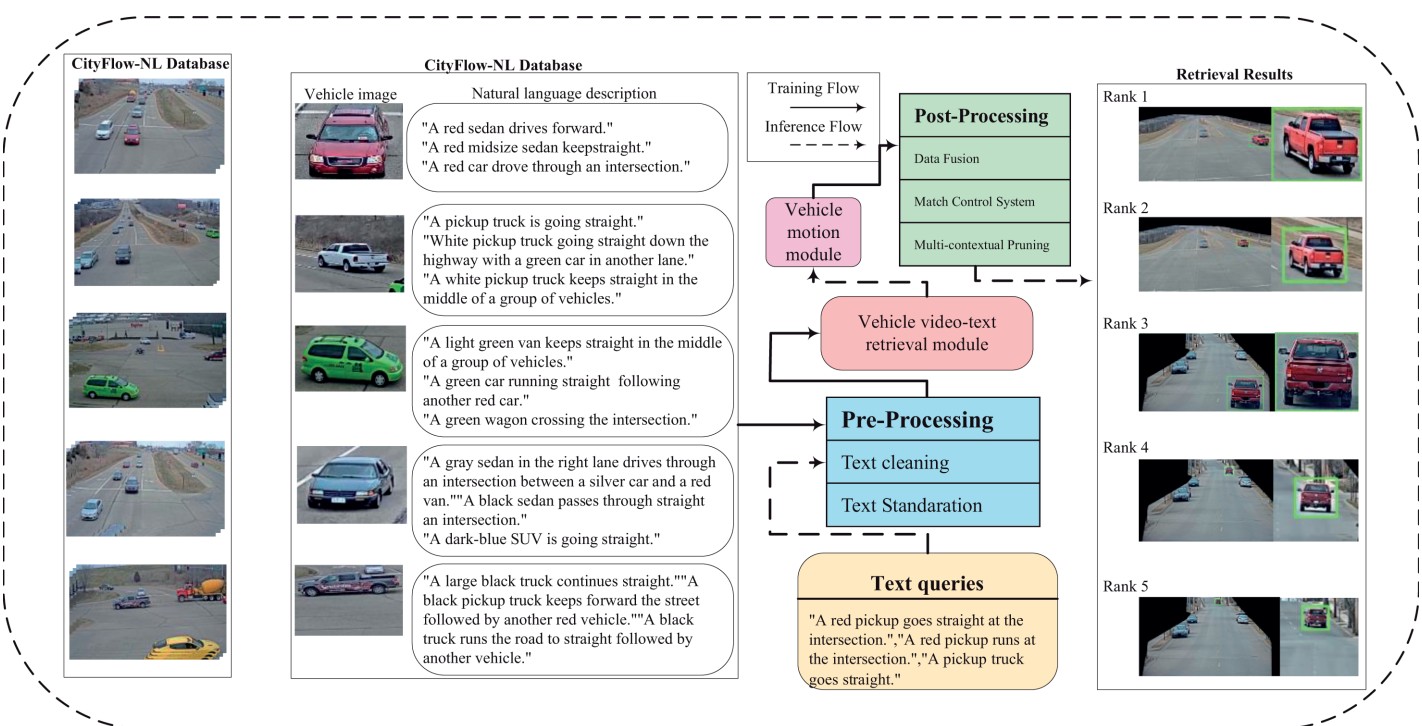

**Fig 1. Overview of our proposed MVR framework.** During the training phase, following preprocessing, both vehicle images and natural language descriptions are fed into the Vehicle video-text retrieval module, leading to the development of a video-text retrieval model. During the inference phase, the preprocessed text queries are processed using the trained video-text retrieval model in conjunction with the vehicle motion module. After undergoing several post-processing steps, the correct vehicle tracks corresponding to these text queries are retrieved.

learning and has been widely applied across a range of natural language processing and computer vision tasks. For instance, in the context of visual recognition, the contextual Transformer (CoT) block [25] is designed to effectively enhance visual recognition capabilities. Additionally, Agent attention [26] has emerged as a formidable competitor to the visual Transformer's multi-head attention, seamlessly combining potent Softmax attention with efficient linear attention and posing a significant challenge to the established approach. Extensive experiments have demonstrated the effectiveness of agent attention in various Vision Transformers [27] and diverse visual tasks. Concurrently, studies have highlighted the potential of integrating pre-processing networks (e.g., ResNet) with ViT [28] among the variants of ViT.

Building upon these advancements, our multimodal language vehicle retrieval (MVR) system integrates video and language information, leveraging contrastive learning to extract vehicle image, frame, and text attributes. This approach aims to develop a robust and effective vehicle video retrieval method.

## Video retrieval by natural language descriptions

In the field of video retrieval, numerous algorithms have been developed, based on multimodal and contrastive learning principles. For example, CLIP4CLIP [29] specifically adapts the CLIP model for video retrieval tasks. It extends the contrastive learning approach by adopting various techniques to combine the features of frames with the textual description of the video. This adjustment enriches the model's ability to obtain comprehensive semantic representations from video and text data, thereby improving its retrieval accuracy.

Another notable algorithm, X-CLIP [30], combines the CLIP model with the Transformer architecture to promote cross-modal learning. By effectively combining visual, temporal, and textual information, X-CLIP performs well in various video retrieval tasks. In addition, algorithms such as Cap4Video [31], CenterCLIP [32], CLIP2Video [33], VOP [34], T-MASS[35], and CLIP2TV [36] have also made significant progress in improving video retrieval capabilities. Moreover, Che et al. [37] have proposed a method for video retrieval that emphasizes video-text alignment, seeking to bridge the inherent semantic gap between these two modalities. Building on this, Li et al. [38] introduced the CMFI module, which tackles the problem of frame underexpression in videos by acknowledging that audio often provides supplementary or essential information for text-to-video retrieval tasks.

Inspired by the success of these algorithms, our MVR system aims to implement a robust and effective vehicle video retrieval approach. By integrating video and language information, and by adopting contrastive learning techniques, our system extracts meaningful features from vehicle frames and attributes, thereby improving the accuracy and effectiveness of vehicle video retrieval using natural language.

## Method

### Method overview

In this section, we discuss the main components of the multi-modal vehicle retrieval (MVR) system, which is based on video-text retrieval. MVR combines multiple techniques and strategies to improve vehicle retrieval by adopting multimodal information such as text, image, and video. The system focuses on gaining a comprehensive understanding of vehicle information and trajectories by integrating various data types. It extracts effective features and deep understanding from useful data by combining multiple deep learning models, post-processing methods, and optimization algorithms.

The proposed MVR system, as shown in Fig 1, consists of four main modules: (1) pre-processing, (2) baseline vehicle video-text retrieval module, (3) vehicle motion module, (4) post-processing module. Each module handles the task at a different level of detail, ultimately achieving superior performance on benchmark datasets. By integrating these four modules, the MVR model has been proven to produce reliable results, promoting the advancement of vehicle retrieval methods in the field of transportation.

### Natural language analysis

In the context of vehicle retrieval using natural language, textual information plays an essential role in offering comprehensive semantic descriptions. By applying statistical analysis and natural language processing techniques to these descriptions, we can accurately identify and extract key attributes such as vehicle color, type, and motion. To facilitate this process, we use a keyword parser that classifies vehicle information based on a predefined set of keywords. These keywords include color categories (e.g., black, white, red, blue, green), vehicle types (e.g., sedan, SUV, pickup, van, bus, truck), and motion directions (e.g., straight, stop, left, right). The comprehensive list of keywords for each category is presented in Table 1.

Our analysis reveals a significant connection between natural language (NL) descriptions and their corresponding descriptions from alternative perspectives, which we refer to as "NL other view descriptions". These descriptions can be based on scenarios different from the primary one. As depicted in Fig 2, there exists a distinct relationship between NL descriptions and NL other view descriptions, hinting at a potential, albeit weaker, connection that allows scenario 1 to be translated into scenario 2 or scenario 3. In the creation of text-video

**Table 1. Vehicle natural language description lables from CityFlow-NL dataset.**

| Class | Labels |
|---|---|
| Color | blue, brown, gray, orange, black, purple, silver, green, white, yellow, red |
| Type | sedan, SUV, pickup, van, bus, truck |
| Motion | straight, stop, left, right |

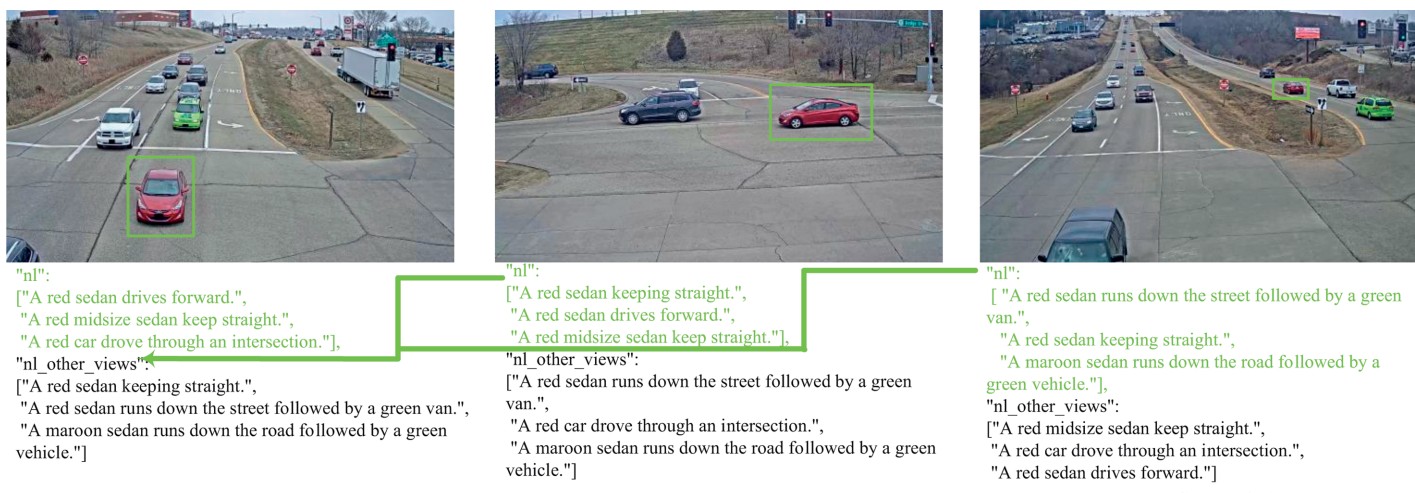

"nl":
["A red sedan drives forward.",
 "A red midsize sedan keep straight.",
 "A red car drove through an intersection."],
"nl_other_views":
["A red sedan keeping straight.",
 "A red sedan runs down the street followed by a green van.",
 "A maroon sedan runs down the road followed by a green vehicle."]

**Scenario1**

"nl":
["A red sedan keeping straight.",
 "A red sedan drives forward.",
 "A red midsize sedan keep straight."],
"nl_other_views":
["A red sedan runs down the street followed by a green van.",
 "A red car drove through an intersection.",
 "A maroon sedan runs down the road followed by a green vehicle."]

**Scenario2**

"nl":
 [ "A red sedan runs down the street followed by a green van.",
   "A red sedan keeping straight.",
   "A maroon sedan runs down the road followed by a green vehicle."],
"nl_other_views":
["A red midsize sedan keep straight.",
 "A red car drove through an intersection.",
 "A red sedan drives forward."]

**Scenario3**

**Fig 2. Different video frames and natural language (NL) descriptions of the same vehicle are included in the CityFlow-NL training set.** In the natural language description, information such as the color, type, and motion trajectory of the vehicle is provided, as shown in the figure. Some NL descriptions from other camera views are placed in the natural language other views.

pairs, we assign a partial penalty weight to NL descriptions from other views, as compared to standard NL descriptions. This approach clarifies the relationship between the primary and supplementary scenes.

## Frame analysis

Video frame information is a key component of the text-to-video module and has a significant impact on the retrieval process. Each camera video is divided into multiple video clips. To facilitate the tracking and identification of objects within these clips, we employ a tracking algorithm, such as DeepSORT, which automatically assigns a unique track ID to each object and generates a corresponding bounding box around it in each frame of the video clips. This process ensures accurate and efficient object tracking throughout the video retrieval process.

In the video recognition model, a clean background of the local road is generated by computing the median value of each pixel across all frames in the video. Utilizing this background, a brief video clip can be created, which incorporates a mask for the region of interest (ROI) and highlights the vehicle corresponding to the specified track ID. To further enhance image quality and boost the model's robustness, a random number is chosen to dictate the frame interval for each iteration. An illustration of a processed video frame is provided in the left corner of Fig 3.

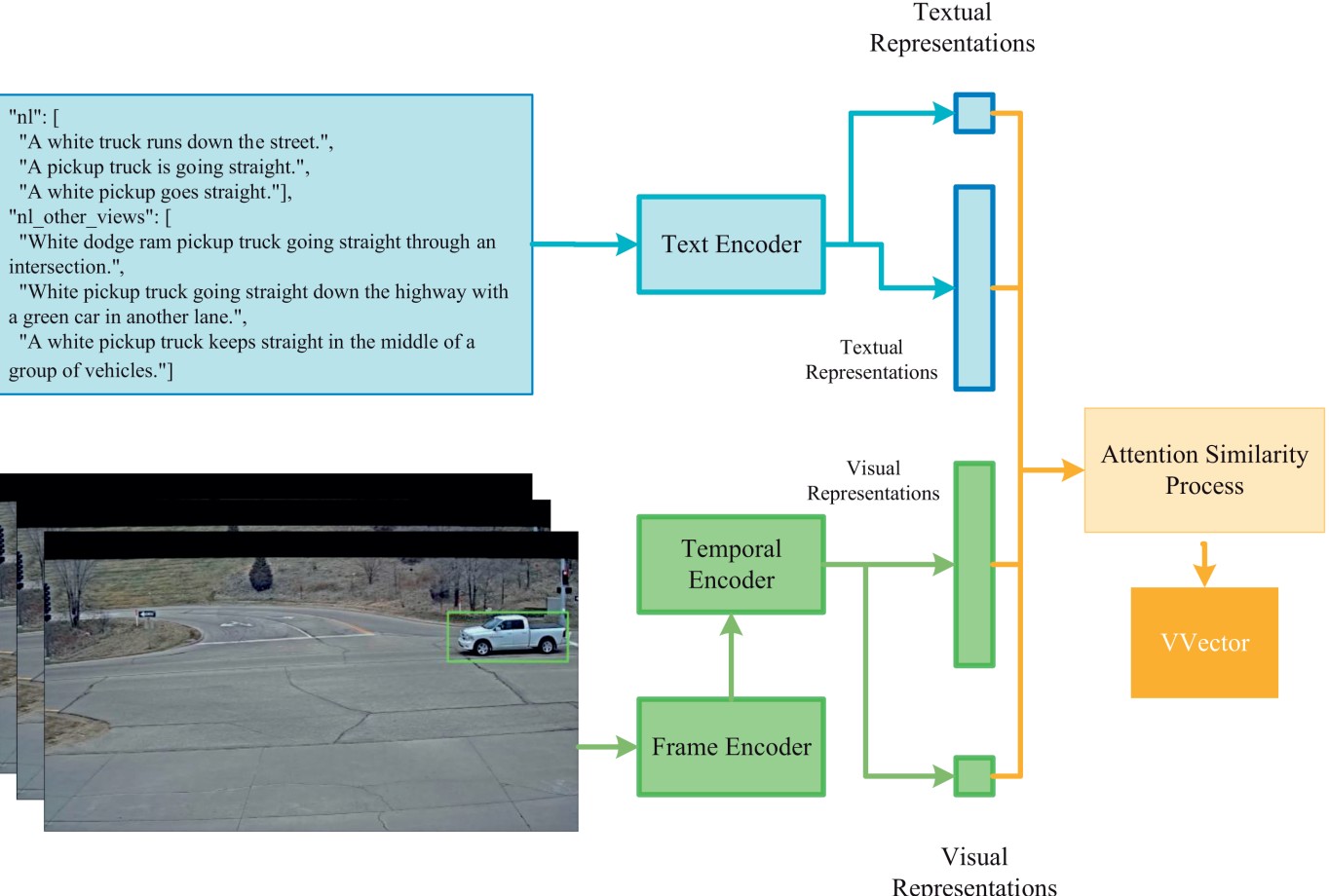

**Fig 3. The core architecture of our video recognition module involves different processing steps for visual and textual data.** Video frames are fed into a frame encoder(Vision Transformer) and a temporal encoder to generate visual representations. Meanwhile, textual information is processed through a text encoder(Text Transformer [39]) to generate corresponding textual representations. These visual and textual representations are then combined via an attention similarity mechanism to predict a score for model evaluation and updating.

## Pre-processing

The inconsistency in text data within the retrieval model can impair the model's performance. To address this, we implemented common text cleaning techniques in natural language processing: stemming and lemmatization. Stemming reduces words to their root forms, while lemmatization restores them to their base or dictionary forms. This process, which includes removing stop words, correcting misspelled words, and converting words to their consistent basic forms, enhances the model's generalization capability during the learning process.

**Text cleaning**. From the perspective of language and grammar, we use stemming to truncate the ends of words to obtain their root forms, and we use Lemmatization to restore words to their basic forms. This process includes stop word removal, correcting misspelled words, and converting words to basic forms for consistent text formatting.

**Text standardization**. To address semantic inconsistency and reduce variation in text embeddings, we standardize the text descriptions. Based on the dataset provided by the AI City Challenge, a natural language description of a vehicle typically comprises three parts: color, type, and trajectory, as outlined in Table 1. By employing semantic role labeling

(SRL) [40], we extract relevant information and standardize the query into the following format: $t_{Standardized} = a_c + a_t + a_m, a_c, a_t, a_m$ represent the color, type, and movement of the vehicle, respectively.

To further address the diversity issue in text descriptions, we apply the same semantic similarity threshold principle. Words are clustered based on their semantic similarity, but only if they exceed the threshold. For example, while 'brown' and 'beige' may be considered synonyms, we only cluster them if their similarity score meets or exceeds the threshold. If a word's similarity score falls below the threshold and it is deemed too ambiguous, it remains unchanged. Additionally, to standardize the description of vehicle motion trajectories, we have categorized them into four types: straight, left turn, right turn, and stop, as detailed in Table 1.

## Baseline vehicle video-text retrieval module

The vehicle video-text retrieval module, which serves as the foundation of our MVR model, is adapted from the X-CLIP algorithm. This algorithm effectively discerns the relationship between video clips and their respective text descriptions. Each track ID is assigned multiple natural language descriptions to facilitate the generation of corresponding video-text pairs for model training. A series of experimental results by Ma et al. [30] demonstrate that this method can indeed effectively achieve outstanding performance on five widely-used video-text retrieval datasets. Furthermore, the weights of descriptions from different perspectives within the video-text pair are adjusted to minimize their impact on the matching results.

Fig 3 illustrates the architecture of the vehicle video-text retrieval module. Frames are first fed into the frame encoder to extract visual features, which are then processed by the temporal encoder to construct temporal information. Consequently, visual representations and their corresponding average pooled vectors are produced. Simultaneously, the text encoder processes textual information to generate text representations, encompassing both sentence-level and word-level data. The attention similarity process integrates the visual and text representations to calculate the video vector for model evaluation and optimization. The video vector is shown as follows:

$$V\left(v_i, t_j\right) = \left[S\left(v_i, t_{j1}\right), S\left(v_i, t_{j2}\right), \dots, S\left(v_i, t_{jk}\right)\right], \tag{1}$$

where $V\left(v_i, t_j\right)$ represents the video vector corresponding to video $ID_i$ and $ID_j$. The term $S\left(v_i, t_{jk}\right)$ indicates the score associated with video $ID_i$ and text $ID_j$ of the $k$-$th$ natural language description. In addition, $S\left(v_i, t_{jk}\right)$ can be expressed as:

$$S\left(v_i, t_{jk}\right) = w_{jk} \times \left(S_{vs} + S_{vw} + S_{fs} + S_{fw}\right)/4, \tag{2}$$

where $w_{jk}$ represents the weight assigned to the $k$-$th$ natural language description of text $ID_j$, reflecting its importance in the retrieval task. If this description belongs to the corresponding natural language description, the associated weight will be higher; otherwise, if it represents a different natural language perspective, the weight will be lower. The variables $S_{vs}$, $S_{vw}$, $S_{fs}$, and $S_{fw}$ represent the video-sentence score, video-word score, frame-sentence score and frame-word score, respectively. These scores are generated by the baseline vehicle video-text retrieval module through the following processes:

- Video-Sentence Score $S_{vs}$: This score is calculated by measuring the semantic similarity between the entire video content and the given sentence. Techniques such as

cosine similarity on feature vectors extracted from a pre-trained video-language model can be employed.

- Video-Word Score $S_{vw}$: This score assesses the relevance of individual words in the sentence to the video content. It can be computed by averaging the similarity scores between each word in the sentence and the video features, possibly using word embeddings and video frame embeddings.
- Frame-Sentence Score, $S_{fs}$: Similar to $S_{vs}$, this score measures the semantic similarity between each frame of the video and the entire sentence. This can be achieved by computing the similarity for each frame-sentence pair and then averaging these similarities.
- Frame-Word Score $S_{fw}$: This score evaluates the relevance of individual words to specific frames in the video. It is calculated by averaging the similarity scores between each word and each frame, utilizing word embeddings and frame-level features.

By integrating these scores, the baseline module provides a comprehensive assessment of the video-text relevance, which is then weighted by $w_{jk}$ to reflect the importance of the natural language description in the context of the retrieval task.

The symmetric InfoNCE loss [41] is combined with the average value of the video vector to optimize the baseline vehicle video-text retrieval module. This loss function comprises the video-to-text loss and the text-to-video loss, which are calculated as follows:

$$L_{v2t} = -\frac{1}{N}\sum_{i=1}^{N}\log\frac{\exp\left(\operatorname{mean}\left(V\left(v_i, t_j\right)\right)\right)}{\sum_{i=1}^{N}\exp\left(\operatorname{mean}\left(V\left(v_i, t_j\right)\right)\right)}, \tag{3}$$

$$L_{t2v} = -\frac{1}{N}\sum_{i=1}^{N}\log\frac{\exp\left(\operatorname{mean}\left(V\left(v_j, t_i\right)\right)\right)}{\sum_{j=1}^{N}\exp\left(\operatorname{mean}\left(V\left(v_j, t_j\right)\right)\right)}, \tag{4}$$

$$L_{vrm} = L_{v2t} + L_{t2v}, \tag{5}$$

where $L_{vrm}$ is the baseline vehicle video-text retrieval module loss, which comprises the video-to-text loss $L_{v2t}$ and the text-to-video loss $L_{t2v}$. By incorporating $L_{t2v}$, this method not only improves the overall performance of the MVR system, but also enhances its applicability and generalization ability in video-text retrieval.

## Vehicle motion module

This module is dedicated to analyzing vehicle maneuvering trajectories with the aim of developing advanced directional control systems. By utilizing the given trajectory, which comprises a series of bounding boxes, we derive the starting vector of the vehicle by applying linear regression to the initial segment of the trajectory's center points. Similarly, the ending vector is obtained by fitting the center points of the final segment using another linear regression function. The angle between these two vectors is then calculated to establish a reference point. To determine the final direction, a set of threshold parameters is assumed.

These threshold parameters are chosen through an empirical testing process combined with optimization techniques. Initial threshold values are selected based on domain knowledge and preliminary experiments. Subsequently, these thresholds are fine-tuned using cross-validation methods on a labeled dataset containing various vehicle maneuvers. Specifically, we use k-fold cross-validation to evaluate different threshold settings and choose the one that minimizes classification error while maintaining robustness across different types of vehicle

movements. For example, if the angle falls between -10 and 5 degrees, the vehicle is classified as moving straight. The process is illustrated in an intuitive manner in Fig 4, which clearly outlines the approach we adopt in the vehicle motion module.

## Post-processing

**Data fusion.** In this subsection, we focus on the post-processing and result fusion of the MVR model. The model generates vector outputs from two different modules, which are then weighted and fused to create a comprehensive vector. The weighted vector is denoted as $V^w$ and is calculated using the following formula:

$$V^w = \left( w_1^w \times \sum_{i=1}^{3} V_i^v + w_2^w \times \max_{i=1}^{n} V_i^v \right) \times \frac{1}{2},$$ (6)

where $V_i^v$ represents the *i-th* text-video pair vector, while $w_1^w$ and $w_2^w$ are the weights associated with the natural language (NL) description and all descriptions, respectively. These weights $w_1^w$ and $w_2^w$ are manually specified and their specific values have been predetermined. To ensure reproducibility, the exact values of $w_1^w$ and $w_2^w$ are set to $\frac{1}{2}$ and $\frac{1}{2}$ , respectively. The factor of $\frac{1}{2}$ in Eq (6) serves as a simple averaging mechanism to balance the contributions from the weighted sum of the first three NL text-video pairs and the weighted maximum vector among all pairs. This approach aims to capture both the general trend from the primary NL descriptions and the most salient information from all descriptions in a straightforward and intuitive manner. To extract the primary information from the NL description, the first three NL text-video pairs are averaged. Additionally, the maximum vector within the entire set of text-video pairs is deemed to encapsulate information from other natural language descriptions. The fused vector, which integrates vector information from two modules, is denoted as $V^f$ and is calculated using the following formula:

$$V^f = \left( w_m^f \times V^m + w_v^f \times V^v \right),$$ (7)

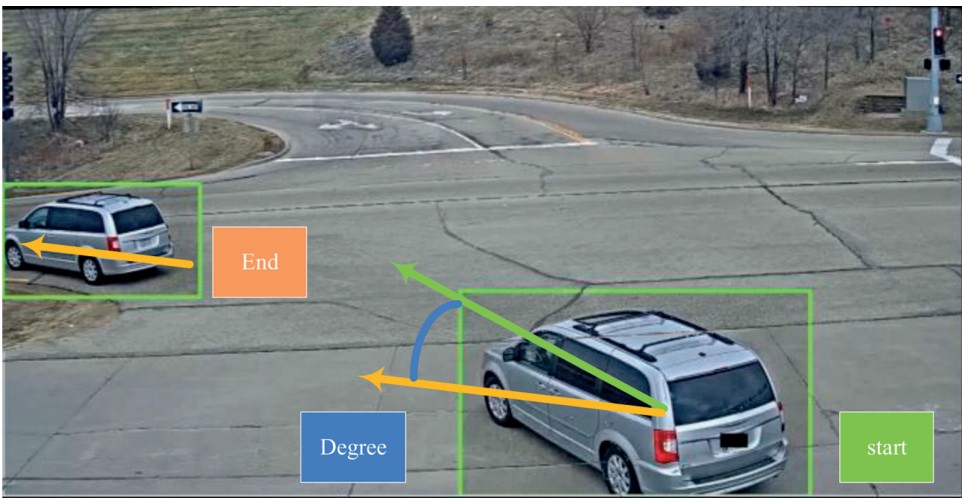

**Fig 4. Example of a vehicle motion module.** In the figure, the starting vector is represented by a green arrow, the ending vector is represented by an orange arrow, and the angle between the two vectors is represented by a blue arc.

where $V^m$ and $V^v$ represent MVector and VVector, respectively, which are generated by the baseline vehicle video-text retrieval module and the vehicle motion module. The weights $w_m^f$ and $w_v^f$ are manually specified with predetermined values to effectively integrate information from these modules. To ensure reproducibility, the exact values of $w_m^f$ and $w_v^f$ are set to $\frac{1}{2}$ and $\frac{1}{2}$, respectively. This systematic approach to vector computation ensures that the MVR model effectively integrates information from a variety of sources, thereby enhancing the overall rigor and robustness of the retrieval process.

**Match control system.** After data fusion, we employ the matching control system from Reference [7] to ascertain the optimal text-video pairings. This system serves to re-rank the initial score matrix, enhancing the ultimate mean reciprocal rank (MRR) metric. Algorithm 1 delineates the fundamental procedures of the Match Control System. An input text-video matrix is constructed with dimensions $m \times n$, where $m$ represents the number of unique text IDs and $n$ denotes the number of distinct video IDs. Within the algorithmic flow, for every row in the input matrix, the highest scoring column index ($hci$) and the highest scoring row index ($hri$) are identified. In instances where the current row under consideration corresponds to the highest scoring row index, a consistent threshold value ($th$), predetermined and invariant throughout all experiments, is subtracted from each entry in column $hci$, with the exception of the element situated at position $tv[hri, hci]$.

The threshold $th$ used in Algorithm 1 (Match Control System) is indeed a predefined value that remains constant across our experiments. It is not dataset-specific but rather a fixed parameter chosen based on empirical observations and validation during the development of our method. This ensures consistency in the application of the matching control system across different datasets, allowing for a fair comparison of results.

**Algorithm 1. Match control system.**

**Input:** Text-video matrix $m$
1 **for** *each row in $m$* **do**
2 Get the highest score column index $I_{ci}$ in $m[\text{row},:]$
3 Get the highest score row index $I_{ri}$ in $m[:, I_{ci}]$
4 **if** *row = $I_{ri}$* **then**
5 **for** *every item in column $I_{ci}$ except $m[I_{ri}, I_{ci}]$* **do**
6 Minus a threshold $th$
7 **end**
8 **end**

Fig 5 illustrates an example of a $7 \times 7$ text-video matrix. In this example, the highest score is located at the intersection of row 2 and column 4, indicating that text ID 3 and video ID 5 are the best match. Consequently, a predetermined threshold deviation is subtracted from all other elements in column 4. This process is iteratively applied iteratively to all rows, leading to a more robust and efficient matching procedure.

**Multi-context pruning.** Based on our observation, imposing strict constraints on the contextual attributes of vehicles can significantly enhance the accuracy of the final results. We employ a multi-context pruning method [3] that eliminates tracks with attributes that deviate from those specified in the description, such as color, type, and motion. In contrast to our previous unidirectional approach, this method incorporates a novel motion analysis module to prune tracks based on bidirectional information, considering multiple directions of the vehicle's movement. This integration enables us to more precisely identify and discard mismatched vehicle tracks, thereby improving the overall model accuracy.

First stage pruning: We evaluate the contextual attributes of each vehicle in the current ranking, sequentially examining the type ($\theta_t$) and color ($\theta_c$). Tracks that align with the

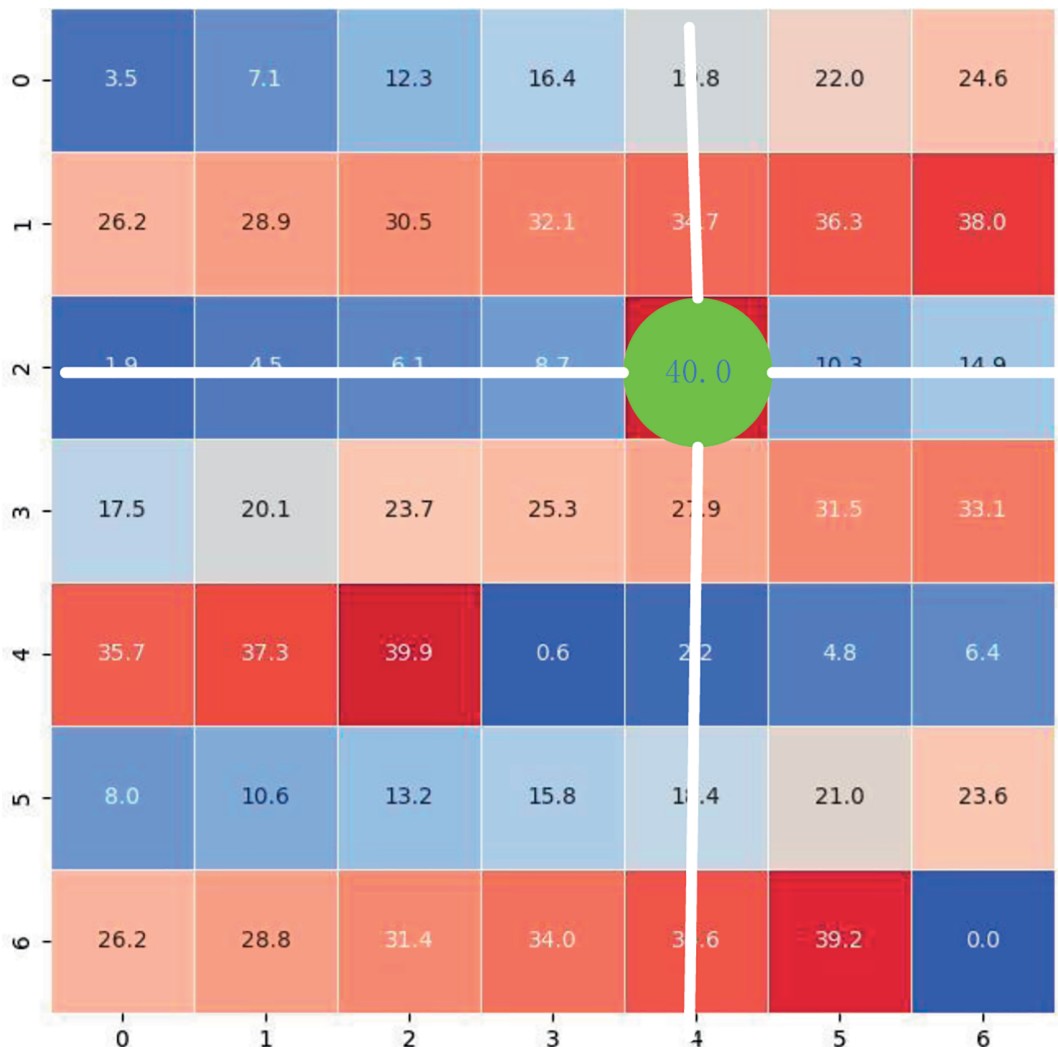

**Fig 5. The example of control match system.**

description are retained in the priority list, while others are demoted. Consequently, the remaining tracks in the priority list will share the same type and color as as specified in the description.

Second stage pruning: Tracks in the priority list are reorganized based on their primary direction ($\theta_d$). Given that the description may encompass multiple directions, we predict all plausible directions for each and then reorder the priority list accordingly. The overall pruning process is outlined in Algorithm 2.

## Experiment

### Dataset

This work utilizes the CityFlow-NL benchmark dataset [42], which is tailored for multimodal trajectory description and understanding tasks. The dataset encompasses 10 intersections in

**Algorithm 2.  Multi-contextual pruning.**

**Input:** `text, current_rank`
**Output:** `pruned list`

```
 1  Initialize empty lists;
 2  begin
 3  │    irrelevant_list ← {};
 4  │    priority_list ← {};
 5  end
 6  for each u in current_rank do
 7  │    if get_type(u) == get_type(text) and get_color(u) == get_color(text)  then
 8  │    │    Add u to priority_list;
 9  │    end
10  │    else
11  │    │    Add u to irrelevant_list;
12  │    end
13  end
14  Initialize empty lists;
15  begin
16  │    highly_relevant ← {};
17  │    likely_relevant ← {};
18  │    moderately_relevant ← {};
19  end
20  Get directions from text;
21  begin
22  │    directions ← get_directions(text);
23  end
24  for each u in priority_list do
25  │    Get directions from u;
26  │    begin
27  │    │    u_directions ← get_directions(u);
28  │    end
29  │    if match_all (u_directions, directions, tolerance=5°)  then
30  │    │    Add u to highly_relevant;
31  │    end
32  │    else if partial_match(u_directions, directions, threshold=0.7)  then
33  │    │    Add u to moderately_relevant;
34  │    end
35  │    else
36  │    │    Add u to likely_relevant;
37  │    end
38  end
39  Return highly_relevant + likely_relevant + moderately_relevant +
       irrelevant_list;
```

a medium-sized city in the United States, along with 3.25 hours of video captured by 40 cameras. It comprises 2,155 vehicle trajectories, each accompanied by three corresponding natural language descriptions, providing a rich and diverse set of scenarios for analysis. The maximum distance between two cameras in the same scene is 4 km, covering a wide range of locations including intersections, stretches of roadways, and highways. This spatial diversity contributes to the dataset's ability to capture a broad spectrum of traffic conditions and patterns. Furthermore, 184 unique vehicle trajectories have been extracted from the original dataset to form a test set, enabling a robust assessment of the model's performance.

The 3.25-hour video dataset comprises footage aggregated from multiple recording sessions conducted across different days, capturing seasonal variations (e.g., winter scenes with snow piles) and diverse traffic conditions. While the exact collection dates and intervals are not specified due to privacy constraints, the dataset's broad spatial coverage—spanning 10 intersections and 40 cameras—ensures representation of varied environmental and traffic scenarios. To better understand the dataset, we undertook an exhaustive literature review [3,42–44], which facilitated the identification of the primary types of intersections

encompassed within the dataset. These intersections encompass a wide array of urban scenarios, specifically:

- Highways: Distinguished by high-speed vehicular traffic and the presence of multiple lanes catering to vehicles traveling in diverse directions.
- Residential Areas: Typically characterized by slower-moving traffic and narrower streets with reduced vehicular activity.
- City Streets: Exhibiting a heterogeneous mix of traffic flow, pedestrian activity, and varying street widths.

Illustrative examples of video frame images extracted from the Cityflow-nl dataset are presented in Fig 6.

## Evaluation metrics

The task of retrieving vehicles from natural language descriptions is evaluated using standard metrics for retrieval tasks. The mean reciprocal rank (MRR) is used, and the formula is

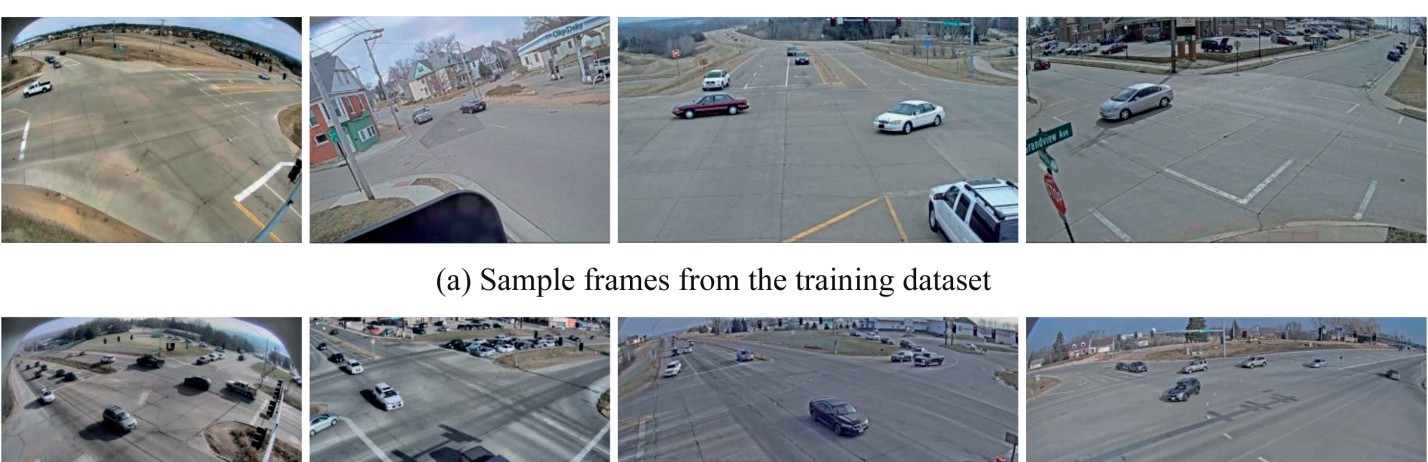

(a) Sample frames from the training dataset

(b) Sample frames from the test dataset

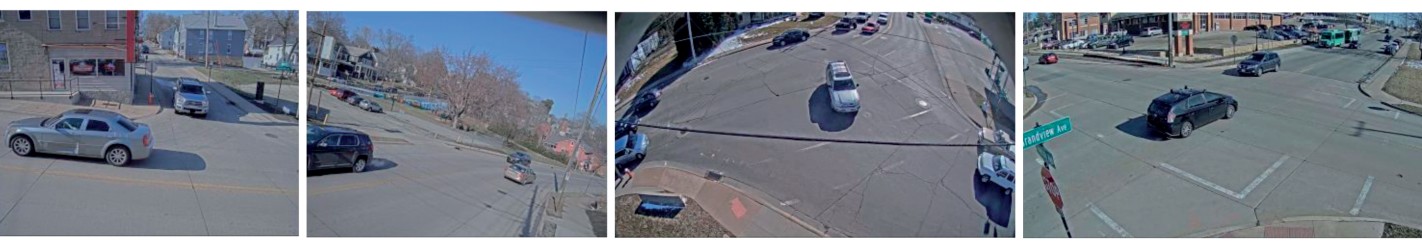

(c) Sample frames from the val dataset

**Fig 6. Examples of video frame images extracted from the Cityflow-nl dataset.**

as follows:

$$MRR = \frac{1}{|Q|} \sum_{i=1}^{|Q|} \frac{1}{rank_i}, \tag{8}$$

where $rank_i$ refers to the ranking position of the correct track for the $i\text{-}th$ text description, and $Q$ is the set of all text queries used in the evaluation. In addition, Recall@5 and Recall@10 are also evaluated.

## Experiments details

All training images are resized to 224×224 pixels and normalized. We adapt the video text retrieval model, X-Clip, as the baseline for the video recognition module. This module incorporates a pre-trained model founded on CLIP, specifically the ViT base architecture with a patch size of 16. The training procedure employs a learning rate of 1e-4 and is configured to accept a maximum of 32 words per text query and a maximum of 20 frames per video clip. The model is trained for 50–60 epochs utilizing a batch size of 64. Our experimental setup adopts a distributed training method and methodology, leveraging four NVIDIA 2080Ti GPUs for efficient parallel computation. To generate the attributes of each vehicle tracklet, we perform inference on all frames of each tracklet, subsequently selecting the class with the highest frequency of occurrences.

## Ablation study

We present a comprehensive analysis of the performance of our MVR model and conduct an ablation study to evaluate the contribution of each module to the overall performance. These results highlight the competitive performance of our MVR model in the context of tracked vehicle retrieval using natural language descriptions.

To critically evaluate the components contributing to the effectiveness of our approach for retrieving tracked vehicles via natural language descriptions, we conducted an ablation study as detailed in Table 2. These tests systematically isolate the impact of individual elements within our MVR framework.

Starting with a baseline model, we observed an initial Mean Reciprocal Rank (MRR) of 0.2562, indicating a modest level of precision in matching textual queries to relevant vehicle records. Upon integrating the Vehicle Motion Module (VMM), which captures dynamic aspects of tracked vehicles, there was a notable improvement, raising the MRR by 0.1250 to 0.3812. Further advancements were achieved by incorporating the Match Control System (MCS). This addition led to an MRR improvement of 0.0268, resulting in a more refined MRR of 0.4080, emphasizing the system's ability to better discern between similar vehicle trajectories.

Ultimately, to maximize the benefits of our pruning strategy (a technique aimed at eliminating irrelevant search candidates in the process), we employed a Multi-Context Pruning

**Table 2. Ablation study analysis of our MVR method. MRR_I: Improvement in MRR, F_MRR: Final MRR Value.**

| Baseline | +VMM | +MCS | +1st Prune | +2nd Prune | MRR_I | F_MRR |
|---|---|---|---|---|---|---|
| ✓ | | | | | - | 0.2562 |
| ✓ | ✓ | | | | +0.1250 | 0.3812 |
| ✓ | ✓ | ✓ | | | +0.0268 | 0.4080 |
| ✓ | ✓ | ✓ | ✓ | | +0.2604 | 0.6684 |
| ✓ | ✓ | ✓ | ✓ | ✓ | +0.2282 | 0.8966 |

Method, leveraging bidirectional attributes. The first stage of pruning contributed an MRR improvement of 0.2604, bringing the MRR to 0.6684. The second stage of pruning further enhanced the MRR by 0.2282, culminating in an impressive peak MRR of 0.8966. This revised table clearly differentiates the contributions of each module, highlighting the effectiveness of each component in enhancing the overall performance of our MVR model.

## Challenge results

Table 3 presents the public leaderboard for the task of retrieving tracked vehicles using natural language descriptions. The table below lists the top 5 results, ranked by their MRR score. Our MVR model surpasses the best-performing team on the 2023 public leaderboard, achieving an MRR score of 0.8966, compared to their score of 0.8263. These results highlight the competitive performance of our MVR model in the application domain of tracking vehicle trajectory retrieval based on natural language descriptions.

## Qualitative analysis

To qualitatively validate the effectiveness of our proposed MVR system, we present some typical text-to-video retrieval examples in Fig 7. From these retrieval results, we find that MVR system can accurately understand the content of sentences and videos. Meanwhile, it is robust in comprehending complex and similar sentences and videos, which is primarily attributed to the multi-grained contrast of our proposed model. Specifically, as shown in the first example in Fig 7, even though the videos are similar, our proposed MVR can still select the correct videos by understanding the details of both sentences and videos. Due to the multi-grained contrast, MVR excels in visual and textual content understanding, enabling it to retrieve the correct vehicle track using natural language descriptions.

## Conclusion

This paper presents Multimodal Vehicle Retrieval (MVR), a novel technical solution for the 2023 AI City Challenge, designed to retrieve vehicle trajectories from natural language descriptions. By employing CLIP for feature extraction, computing weighted fusion vectors, and utilizing matching control systems and post-processing techniques based on multi-context attribute information, MVR effectively integrates text, image, and video data to accurately identify target vehicles in multi-camera surveillance scenarios. Our proposed method achieved a notable mean reciprocal ranking (MRR) of 0.8966 in the second track of the 7th AI City Challenge, underscoring the potential of MVR for multimodal retrieval in transportation applications.

**Table 3. Leaderboard for the tracked vehicle retrieval challenge using natural language descriptions.**

| Rank | Model | MRR |
|---|---|---|
| 1 | **MVR(ours)** | 0.8966 |
| 2 | HCMIU-CVIP [3] | 0.8263 |
| 3 | MLVR [7] | 0.8179 |
| 4 | AIO-NLRetrieve [45] | 0.4795 |
| 5 | AIO2022 [45] | 0.4659 |

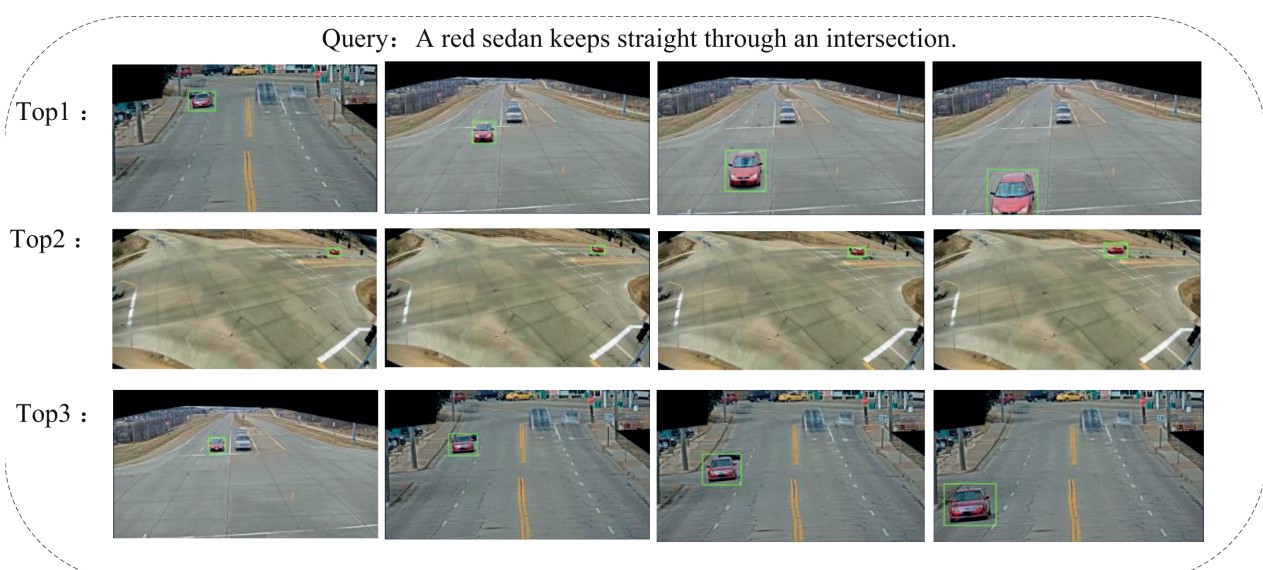

**Fig 7. Top-3 video-to-text retrieval results on CityFlow-NL dataset.** The four images from left to right represent snapshots of the vehicle trajectory over time, corresponding to the text description.

## Limitations

Despite the strong performance of Multimodal Vehicle Retrieval (MVR), the model's generalization is limited by the current dataset, which primarily consists of daytime and fair-weather samples. Real-world deployment demands robustness against adverse conditions such as low illumination and meteorological interferences like rain and fog. Therefore, validating the system's performance in challenging scenarios, including nighttime and inclement weather, remains essential, as these conditions can hinder visual feature extraction and trajectory tracking.

## Future research directions

To address these limitations, future work will focus on two main directions: expanding the dataset to include diverse weather conditions and nighttime scenarios, thereby enhancing the system's robustness; and developing synthetic-to-real domain adaptation techniques using physically realistic weather rendering to enable all-weather video retrieval. This will establish MVR as a foundational tool for smart city surveillance systems.

## Acknowledgments

This work was mainly supported by the National Natural Science Foundation of China (Grant No. 62162061 and Grant No. 62262066). This study was also supported by the Doctoral Research Foundation of Xinjiang Normal University (Grant No. XJNUBS2115), and the Xinjiang Normal University Youth Top Talents Project (Grant No. XJNUQB2022-21). In addition, this study was also supported by the Xinjiang Key Research and Development Program (2022B01007-1).

## Author contributions

**Formal analysis:** Nan Ding.

**Funding acquisition:** Zhandong Liu, Yong Li, Nan Ding.

**Investigation:** Ke Li.

**Methodology:** Zhandong Liu, Nan Ding.

**Project administration:** Yong Li.

**Supervision:** Zhandong Liu.

**Validation:** Ke Li, Xiangwei Qi.

**Visualization:** Xiangwei Qi.

**Writing – original draft:** Changhao Zhang.

**Writing – review & editing:** Zhandong Liu.

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
