## [Decision Letter · Decision Letter 0]

16 Dec 2024

PONE-D-24-43459A Novel Multi-modal Retrieval Framework for Tracking Vehicles Using Natural Language DescriptionsPLOS ONE

Dear Dr. Liu,

Thank you for submitting your manuscript to PLOS ONE. After careful consideration, we feel that it has merit but does not fully meet PLOS ONE’s publication criteria as it currently stands. Therefore, we invite you to submit a revised version of the manuscript that addresses the points raised during the review process.

The manuscript has been reviewed by two experts in machine learning and intelligent transportation systems. Concerns have been consistently raised regarding the validity of the result and the clarity of the methodology. I recommend the authors to revise the manuscript thoroughly to fully address both reviewers comments. 

We look forward to receiving your revised manuscript.

Kind regards,

Zhixia Li, Ph.D.

Academic Editor

PLOS ONE

“1)National Natural Science Foundation of China (Grant No. 62162061 and Grant No. 62262066)

2)Xinjiang Normal University Doctoral Initiation Fund Project (Grant No. XJNUBS2115)

3)Xinjiang Normal University Youth Top Talents Project (Grant No. XJNUQB2022-21).

4)Xinjiang Key Research and Development Program (2022B01007-1)”

“This study was mainly supported by the National Natural Science Foundation of China (Grant No. 62162061 and Grant No. 62262066). This study was also supported by the Xinjiang Normal University Doctoral Initiation Fund Project (Grant No. XJNUBS2115), and the Xinjiang Normal University Youth Top Talents Project (Grant No. XJNUQB2022-21). In addition, this study was also supported by the Xinjiang Key Research and Development Program (2022B01007-1).”

“1)National Natural Science Foundation of China (Grant No. 62162061 and Grant No. 62262066)

2)Xinjiang Normal University Doctoral Initiation Fund Project (Grant No. XJNUBS2115)

3)Xinjiang Normal University Youth Top Talents Project (Grant No. XJNUQB2022-21).

4)Xinjiang Key Research and Development Program (2022B01007-1)”

5. Please amend the manuscript submission data (via Edit Submission) to include authors Changhao Zhang,, Ke Li, Yong Li, Xiangwei Qi, and Nan Ding.

Additional Editor Comments:

The manuscript has been reviewed by two experts in machine learning and intelligent transportation systems. Concerns have been consistently raised regarding the validity of the result and the clarity of the methodology. I recommend the authors to revise the manuscript thoroughly to fully address both reviewers comments.

Reviewers' comments:

Reviewer's Responses to Questions

**Comments to the Author**

1. Is the manuscript technically sound, and do the data support the conclusions?

Reviewer #1: Partly

Reviewer #2: Yes

2. Has the statistical analysis been performed appropriately and rigorously? 

Reviewer #1: No

Reviewer #2: Yes

3. Have the authors made all data underlying the findings in their manuscript fully available?

Reviewer #1: No

Reviewer #2: Yes

4. Is the manuscript presented in an intelligible fashion and written in standard English?

Reviewer #1: Yes

Reviewer #2: Yes

5. Review Comments to the Author

Reviewer #1: This paper addresses an important challenge in multimodal vehicle retrieval using natural language descriptions. It integrates multiple modules, including a baseline video-text retrieval module, a vehicle motion module, a multi-context pruning method, and post-processing, into a coherent framework. The proposed method demonstrates competitive performance on the CityFlow-NL dataset, achieving a notable MRR improvement. The ablation study highlights the contributions of individual components, and the evaluation metrics align with standard practices. However, several aspects of the methodology remain unclear, such as the generation of track IDs and bounding boxes, the handling of thresholds and weights, and the scalability of the system to larger datasets. Additionally, certain figures and explanations require refinement for clarity and reproducibility.

Specific Comments

1. Section: Introduction, First Paragraph: The term "continuous recording" is vague—does it refer to constant monitoring or uninterrupted data collection?

2. Section: Introduction, First Paragraph: Typographical error. Missing the author(s) before "et al."

3. 3. Section: Frame Analysis, First Paragraph: "track ID and a corresponding bounding box." How are these generated? Are they automatically extracted via a tracking algorithm (e.g. DeepSORT)?

4. Section: Frame Analysis, Figure 3: Figure 3 suffers from incomplete text, making it challenging to interpret. Please ensure all text and visual elements are legible and contained within the image boundaries.

5. Section: Pre-processing, Fourth Paragraph: While clustering synonyms (e.g., "brown" and "beige") reduces diversity, it may oversimplify semantic nuances. How does this clustering handle edge cases or ambiguous words?

6. Section: Baseline Vehicle Video-Text Retrieval Module, Equation 2: The formulas for "*V*(*v*_*i*_, *t*_*j*_)" and "*S*(*v*_*i*_, *t*_*jk*_)" are unclear in terms of practical implementation. For example, how are "*S*_*vs*_," "*S*_*vw*_," "*S*_*fs*_," and "*S*_*fw*_" calculated?

7. Section: Vehicle Motion Module, First Paragraph: The text mentions "a set of threshold parameters" but does not specify how these thresholds are chosen or tuned. Please provide more detail about this process.

8. Section: Post-processing, Equation 6: The formula for "*V*_*w*_" divides by 2, but it’s unclear if this is normalization or an arbitrary scaling factor. If it’s normalization, why was this particular factor chosen instead of another method (e.g., accounting for vector norms)?

9. Section: Post-processing, Equations 6 and 7: The weights are ambiguous. Are these fixed values, heuristically assigned, or learned during training? Please clarify to improve reproducibility.

10. Section: Post-processing, Algorithm 1: The algorithm lacks details on how the threshold "th" is chosen. Is it dataset-specific, or does it remain constant across experiments?

11. Section: Dataset, First Paragraph: While the dataset is well-documented, the size (2,155 trajectories) might be small for a task involving multimodal inputs. Larger datasets or data augmentation could enhance generalizability.

12. Section: Multi-Context Pruning, Algorithm 2: The term "match_all" used in the algorithm for comparing directions is ambiguous. How strict or flexible is this matching? For example: Does a partial match (e.g., one direction out of multiple matches) qualify as "highly relevant"? How are tolerances for directional deviations handled?

13. Section: Ablation Study, Table 2: The first pruning stage improves MRR significantly (+0.2604), while the second stage boosts it substantially more (+0.2282). Why is the second stage so much more impactful? Please elaborate.

Reviewer #2: (1) There are already numerous detection algorithms based on multimodal and contrastive learning. I suggest that you need to systematically summarize their deficiencies. Additionally, could you elucidate the specific aspects in which the method proposed in this study has been improved compared to the current methods?

(2) The MVR system proposed by the author significantly enhances the capability of text-based vehicle retrieval. I am curious, as the author repeatedly emphasizes this as an innovative system, on what existing methods does this system build and what kind and degree of innovation has been implemented? From the full text, it appears that the system combines several existing methods.

(3) It is suggested that the authors add a new section after the Introduction or Related Work, using concise statements to succinctly encapsulate the innovative aspects of the paper.

(4) The results demonstrate that the MVR system proposed by the author performs superiorly. I am very curious about how the proposed method differs from the other methods listed in Table 3? In which aspects has the author made improvements that enhanced the detection performance of the MVR system?

(5) Please explain the role of multi-grained contrast in this system. How does it improve the detection performance of the algorithm?

(6) In the Experimental section, what type of intersections were the selected 10? It is recommended to upload images of the intersections to enable a better understanding for the readers. Moreover, is the data obtained from 3.25 hours of video representative? What was the weather like during sample collection? Was it during the day or at night? Environmental changes are crucial for trajectory recognition.

6. PLOS authors have the option to publish the peer review history of their article (what does this mean?). If published, this will include your full peer review and any attached files.

Reviewer #1: No

Reviewer #2: No

---

## [Author Response · Author response to Decision Letter 1]

14 Feb 2025

Our response to the specific comments from the reviewers and editor has been uploaded to the submission system in the form of a PDF file, with the file name "Response to Reviewers: PONE-D-24-43459.R1.pdf".

---

## [Decision Letter · Decision Letter 1]

25 Mar 2025

PONE-D-24-43459R1A Novel Multi-modal Retrieval Framework for Tracking Vehicles Using Natural Language DescriptionsPLOS ONE

Dear Dr. Liu,

Thank you for submitting your manuscript to PLOS ONE. After careful consideration, we feel that it has merit but does not fully meet PLOS ONE’s publication criteria as it currently stands. Therefore, we invite you to submit a revised version of the manuscript that addresses the points raised during the review process.

Both reviewers are generally satisfied with the R1, while some minor comments still need to be addressed. A minor revision is needed before the manuscript could be published.

We look forward to receiving your revised manuscript.

Kind regards,

Zhixia Li, Ph.D.

Academic Editor

PLOS ONE

Journal Requirements:

Additional Editor Comments:

Both reviewers are generally satisfied with the R1, while some minor comments still need to be addressed. A minor revision is needed before the manuscript could be published.

Reviewers' comments:

Reviewer's Responses to Questions

**Comments to the Author**

1. If the authors have adequately addressed your comments raised in a previous round of review and you feel that this manuscript is now acceptable for publication, you may indicate that here to bypass the “Comments to the Author” section, enter your conflict of interest statement in the “Confidential to Editor” section, and submit your "Accept" recommendation.

Reviewer #1: All comments have been addressed

Reviewer #2: All comments have been addressed

2. Is the manuscript technically sound, and do the data support the conclusions?

Reviewer #1: Partly

Reviewer #2: Yes

3. Has the statistical analysis been performed appropriately and rigorously? 

Reviewer #1: I Don't Know

Reviewer #2: Yes

4. Have the authors made all data underlying the findings in their manuscript fully available?

Reviewer #1: No

Reviewer #2: Yes

5. Is the manuscript presented in an intelligible fashion and written in standard English?

Reviewer #1: Yes

Reviewer #2: Yes

6. Review Comments to the Author

Reviewer #1: (No Response)

Reviewer #2: The author has addressed my concerns well. However, I have two additional suggestions:

(1) While the author explains that the 3.25-hour dataset is sufficiently large, sample size is only one aspect of representativeness. To ensure the sample is truly representative and to avoid randomness, data should be collected from multiple different dates, even if only for one hour each. Were the 3.25 hours of data all from the same day? The author should clarify this point and discuss the representativeness of the data from this perspective.

(2) The author provides reasonable explanations regarding the factors of weather and nighttime. However, I suggest incorporating relevant discussions into the "Limitations" and "Future Research Directions" sections of the conclusion.

7. PLOS authors have the option to publish the peer review history of their article (what does this mean?). If published, this will include your full peer review and any attached files.

Reviewer #1: No

Reviewer #2: No

---

## [Author Response · Author response to Decision Letter 2]

16 Apr 2025

The response to the reviewers' and editors' comments should refer to the file named "Response of PONE-D-24-43459.R2.pdf".

---

## [Decision Letter · Decision Letter 2]

17 Jun 2025

A Novel Multi-modal Retrieval Framework for Tracking Vehicles Using Natural Language Descriptions

PONE-D-24-43459R2

Dear Dr. Liu,

We’re pleased to inform you that your manuscript has been judged scientifically suitable for publication and will be formally accepted for publication once it meets all outstanding technical requirements.

Kind regards,

Zhixia Li, Ph.D.

Academic Editor

PLOS ONE

Additional Editor Comments (optional):

Reviewers' comments:

Reviewer's Responses to Questions

**Comments to the Author**

1. If the authors have adequately addressed your comments raised in a previous round of review and you feel that this manuscript is now acceptable for publication, you may indicate that here to bypass the “Comments to the Author” section, enter your conflict of interest statement in the “Confidential to Editor” section, and submit your "Accept" recommendation.

Reviewer #1: All comments have been addressed

Reviewer #2: All comments have been addressed

2. Is the manuscript technically sound, and do the data support the conclusions?

Reviewer #1: Partly

Reviewer #2: Yes

3. Has the statistical analysis been performed appropriately and rigorously? 

Reviewer #1: N/A

Reviewer #2: Yes

4. Have the authors made all data underlying the findings in their manuscript fully available?

Reviewer #1: No

Reviewer #2: Yes

5. Is the manuscript presented in an intelligible fashion and written in standard English?

Reviewer #1: Yes

Reviewer #2: Yes

6. Review Comments to the Author

Reviewer #1: (No Response)

Reviewer #2: The authors have adequately addressed my comments. I feel that this manuscript is now acceptable for publication.

7. PLOS authors have the option to publish the peer review history of their article (what does this mean?). If published, this will include your full peer review and any attached files.

Reviewer #1: **Yes: **Yifan Xu

Reviewer #2: No
